# The placement of foot-mounted IMU sensors does affect the accuracy of spatial parameters during regular walking

**Arne Küderle[ID]\*, Nils Roth, Jovana Zlatanovic, Markus Zrenner, Bjoern Eskofier, Felix Kluge**

Machine Learning and Data Analytics Lab, Friedrich-Alexander-Universität Erlangen-Nürnberg, Erlangen, Germany

\* arne.kuederle@fau.de

## Abstract

Gait analysis using foot-worn inertial measurement units has proven to be a reliable tool to diagnose and monitor many neurological and musculoskeletal indications. However, only few studies have investigated the robustness of such systems to changes in the sensor attachment and no consensus for suitable sensor positions exists in the research community. Specifically for unsupervised real-world measurements, understanding how the reliability of the monitoring system changes when the sensor is attached differently is from high importance. In these scenarios, placement variations are expected because of user error or personal preferences. In this manuscript, we present the largest study to date comparing different sensor positions and attachments. We recorded 9000 strides with motion-capture reference from 14 healthy participants with six synchronized sensors attached at each foot. Spatial gait parameters were calculated using a double-integration method and compared to the reference system. The results indicate that relevant differences in the accuracy of the stride length exists between the sensor positions. While the average error over multiple strides is comparable, single stride errors and variability parameters differ greatly. We further present a physics model and an analysis of the raw sensor data to understand the origin of the observed differences. This analysis indicates that a variety of attachment parameters can influence the systems' performance. While this is only the starting point to understand and mitigate these types of errors, we conclude that sensor systems and algorithms must be reevaluated when the sensor position or attachment changes.

## Introduction

Gait analysis using wearable inertial measurement units (IMUs) has shown to be a reliable tool to assess motor impairment and disease progression in a variety of neurological and musculoskeletal indications [1–3]. In particular, shoe-worn IMU sensors can measure spatiotemporal gait parameters with high accuracy [4–8]. Changes in these parameters correlate with disease progression [9, 10], medicine intake [11], fall risk [12], and the well-being of the patients [2].

**Data Availability Statement:** The full dataset is available via Zenodo (https://doi.org/10.5281/zenodo.5747173) and a Python library to load and pre-process the data can be obtained from github

(https://github.com/mad-lab-fau/sensor_position_dataset_helper).

**Funding:** This work was supported by the Mobilise-D project that has received funding from the Innovative Medicines Initiative 2 Joint Undertaking (JU) under grant agreement No. 820820. This JU receives support from the European Union's Horizon 2020 research and innovation program and the European Federation of Pharmaceutical Industries and Associations (EFPIA). Content in this publication reflects the authors' view and neither IMI nor the European Union, EFPIA, or any Associated Partners are responsible for any use that may be made of the information contained herein. This work was partially funded by the Bavarian Ministry for Economy, Regional Development & Energy via the Medical Valley Award 2017 (FallRiskPD Project). This work was partially funded by the Deutsche Forschungsgemeinschaft (DFG, German Research Foundation) via the project "Mobility_APP" (grant number 438496663). B. M. Eskofier gratefully acknowledges the support of the German Research Foundation (DFG) within the framework of the Heisenberg professorship program (grant number ES 434/8-1). The funders had no role in study design, data collection and analysis, decision to publish, or preparation of the manuscript.

**Competing interests:** The authors have declared that no competing interests exist.

These sensor units are cheap, small, and relatively easy to use. Hence, they are of high interest for widespread laboratory assessments and unsupervised real world monitoring.

Although various systems based on foot-worn IMUs have been published and made commercially available, widespread consent about certain design elements does not exist. Apart from differences in the hardware of the actual IMU units, sensor setups vary in position and attachment of the sensor units. They are commonly integrated in the shoe at the heel [13], embedded in a removable insole [14, 15], attached with elastic or Velcro band at the instep [5, 6, 16, 17], attached with a clip at the instep [18, 19], or bolted to the lateral side of the shoe [4]. While a multitude of these systems produce reliable results, it remains unclear how robust the systems would be to changes in these parameters. Without this knowledge, findings from individual studies and expected error ranges cannot be compared or applied to new sensor systems that use a different sensor setup.

Dealing with variety in sensor attachment and placement becomes even more important in the context of home monitoring applications. Multiple studies have shown that monitoring patients gait in an unsupervised manner during their everyday life provides valuable complementary information to at-lab measurements [20, 21]. To ensure compliance and usability for long-term recordings, users need to operate the sensor systems independently and ideally, should be allowed to attach the sensors to their own shoes. However, this increases the risk of user error, leading to improper attachment of the sensor units, and naturally increases the placement variability. Further, certain shoes might only support certain attachment modalities, which means that support of multiple sensor position (e.g., lateral over the heel with a clip and at the instep with an elastic band) might be desirable to increase user acceptance. Therefore, a solid understanding on how the performance of gait analysis systems with foot-mounted IMU changes with the position and mode of attachment of the sensor units is required to ensure that reliable gait parameters can be extracted from such unsupervised monitoring scenarios.

Existing research by Anwary et al. [22] already indicates that relevant differences between various sensor positions on the foot exist when it comes to spatial parameters like stride length and overall walking distance. They measured a difference of up to 6 cm in the mean stride length and up to 5 cm/s in the mean gait speed over 15 participants comparing five sensor positions. Their results indicated that placing the sensor on the bony prominence of the first cuboid bone leads to the most reliable gait parameters. However, this conclusion was drawn primarily by comparing the overall calculated walking distance, as this was the only spatial measure with a ground truth reference. Therefore, it is unclear whether an evaluation of stride-level spatial parameters would lead to the same conclusion. Further, all experiments were performed barefoot. Therefore, it remains to be seen if the same results can be obtained with sensors attached to the shoe.

In our own recent work, we attempted to answer this question for running movements [23]. We attached eight IMUs to a pair of running shoes at different positions (four sensors per shoe) and collected data from approximately 2400 strides from 24 healthy participants at various running speeds. A motion capture system was used as reference to provide stride-level ground truth for spatiotemporal parameters. These direct comparisons of multiple sensors revealed an average stride length error of 0.3 cm for the best and an error of −8.6 cm for the worst sensor position. This difference was even larger for individual participants and increased with increasing running speed. The sensor placed in a cavity in the mid-sole of the shoe lead to the best results overall. While the position is impractical in many cases, the firm attachment of the sensor in the cavity appeared to reduce the amount of movement artifacts and vibrations that might negatively affect a position estimation by double integration. However, this study only investigated running motions. Because of the different biomechanics and the occurrence

of higher frequency components in the recorded signal compared to regular gait, it is unclear if these results are transferable to normal walking speeds.

To the best of our knowledge, these two are the only studies attempting a comparison of sensor positions on the foot using real sensor data. However, Tan et al. [24] and Peruzzi et al. [25] made use of motion capture markers to simulate the expected signal of an IMU attached at the position of the marker.

Tan et al. investigated the influence of changes in the exact sensor position at the instep on the calculation of ground reaction forces. They concluded that small deviations in the position only have a minimal influence on the derived ground reaction forces, as long as the orientation of the sensor is kept unchanged. Unfortunately, they did not report on any spatiotemporal gait parameters.

Peruzzi et al. evaluated the validity of a zero-velocity-assumption (ZUPT) during the mid-stance phase at different sensor positions. During a zero-velocity phase, the sensor is assumed to be stationary. This information is used by many state-of-the-art trajectory estimation algorithms to correct for position and velocity drifts [26]. Therefore, only sensor positions that have reliable regions of no movement during a gait cycle are expected to produce reliable spatial parameters. Peruzzi et al. used motion capture markers to simulate IMUs at various positions around the foot and found that sensors on the lateral aspect of the rear foot or on top of the calcaneus are expected to have time phases with the lowest amount of movement compared to all other sensor positions and hence, should result in the most reliable ZUPT updates and the most reliable spatial parameters.

All of these studies show that performance differences depending on the sensor position are expected. This means that the robustness of systems to these changes must be investigated if we cannot ensure that the sensor is always placed at the same exact position. In particular the quantitative results from [22, 23] make it clear that these differences are large enough to affect health and sports applications. Despite the apparent relevance of this topic, position and attachment differences between sensor setups are rarely discussed in literature and no consensus regarding sensor placement exists. Further, an understanding of the exact origins of the reported differences is missing and therefore, no general recommendations or approaches to minimize their influence on calculated parameters exists.

With this paper we contribute to a better fundamental understanding on how position and attachment of the IMUs affect the raw sensor data and the accuracy of spatial parameters (Fig 1). For this we build upon these previous studies, by continuing the works of Anwary et al. [22] and combining it with our experience from our recent study on running [23], but adapting the methodology to regular walking. We compared the stride length calculated from six sensor positions at the shoe with stride-level motion capture reference in around 9,000 strides from healthy participants. Further, by using synchronized sensors we were able to analyze and quantify differences in the raw IMU data directly. We compared these results with a simple physics model to provide a baseline understanding for the potential origins of the observed differences and provide recommendations for future research directions.

## Materials and methods

### Dataset

Fourteen healthy participants (Table 1) performed a variety of walking tests within the motion capture volume of the Fraunhofer IIS L.I.N.K. (localization, identification, navigation, and communication) test center in Nürnberg, Germany. In total this resulted in around 8,989 usable strides with motion capture reference. The full dataset is available via Zenodo (https://

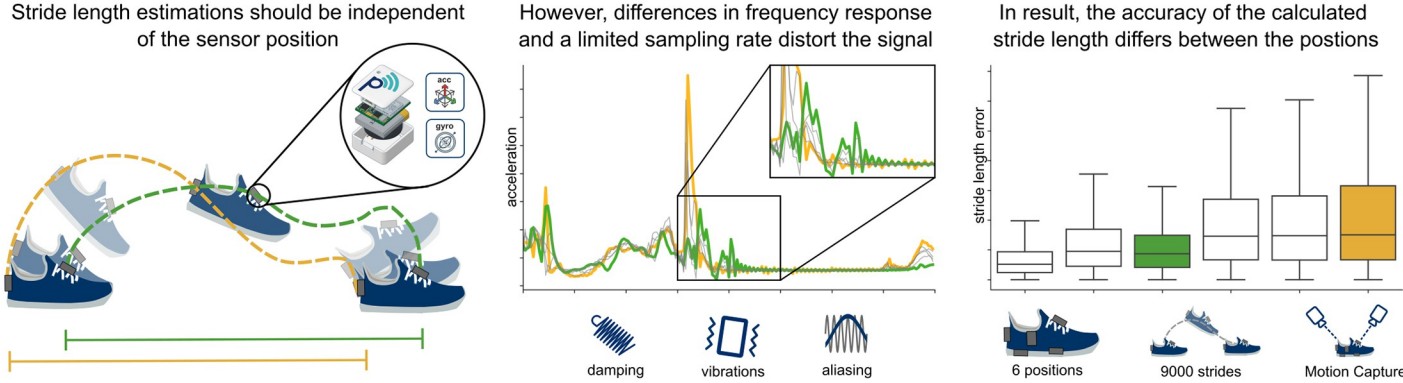

**Fig 1. Visual abstract.** All sensors travel the same distance in the ground-plane over the duration of one stride. Therefore, we expect the calculated stride length to be identical. However, the frequency response of the attachment modulates the recorded signal in a way, that for some sensor positions large integration errors become more likely. We demonstrated this effect with six different sensor positions on a dataset with 9000 strides.

zenodo.org/record/5747173) and a Python library to load and pre-process the data can be obtained from github (https://github.com/mad-lab-fau/sensor_position_dataset_helper).

The participants were each equipped with 13 IMUs (NilsPod v1, Portabiles GmbH, Erlangen, Germany). The IMUs recorded with a sampling rate of 204.8 Hz. The range of the accelerometer was set to ±16 g ($\approx$ ±157 m/s$^2$) and the gyroscope to ±2000 deg/s. Six IMUs were attached to each shoe (Fig 2), one at each ankle, and one at the lower back via a hip belt. For this publication only the sensors attached to the shoes are considered. Two of these sensors required special modifications. One was embedded in the midsole of the purpose-built shoe (Portabiles Healthcare Technology GmBh, Erlangen, Germany), which has a cavity that can fit one of the sensor units. The other was part of a purpose-built insole. It was using the same electronic design as the other sensors, but the housing was modified to fit the size constraints inside the insole (Fig 2, center). The actual IMU in this configuration was placed to be roughly under the center of the foot, depending on the exact size of the foot relative to the size of the insole.

To ensure that differences between the sensor positions are not caused by differences in the sensor units themselves, the positions of the units on the shoe were changed after the fifth and then again after the tenth participant. Only the insole sensors could not be swapped, as only two pairs of the special-build insoles were available for the measurement. For each participant the pair fitting their shoe size best was used. Additionally, all sensor units were manually calibrated according to Ferraris et al. [27] (see [28] for the implementation) to further reduce the sensor to sensor variations.

**Table 1. Participant demographic.**

|  | Demographics |
| --- | --- |
| Gender [f/m] | 4/10 |
| Age [years] | 25.4 ± 3.0 |
| Height [cm] | 178.0 ± 11.0 |
| Mass [kg] | 73.3 ± 15.6 |

All numerical values are provided as *mean ± std*.

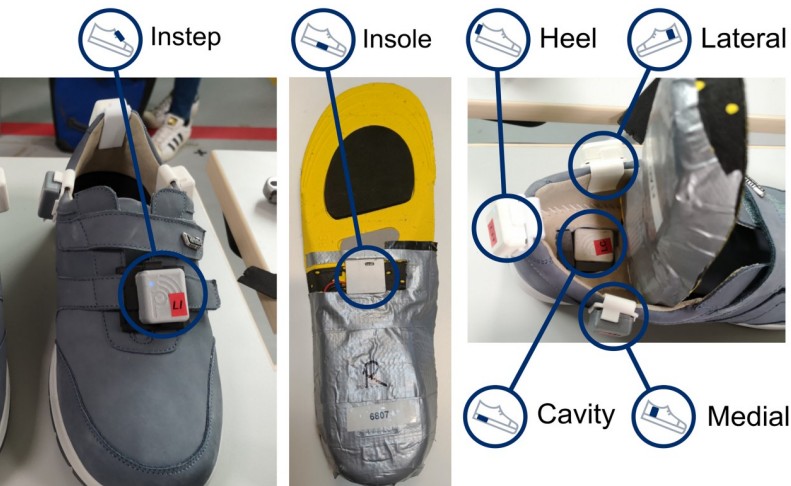

**Fig 2. Sensor placement of the six shoe sensors.** The insole sensor (center) was embedded in a purpose-built insole. The cavity sensor was placed in a cavity in the sole of the custom shoe (right). All other sensors were attached with 3D-printed clips.

For the motion capture system (Opus 700+ Qualisys, 28 cameras, 20×30 m capture volume), a set of eleven reflective markers were attached to the shoes and the hip of each participant [29, 30]. Four markers were attached to each shoe at the calcaneus (CAL), at the tip of the shoe (TOE), and on top of the first and the fifth metatarsal (MET1 and MET5). The remaining three markers were attached at the hip to the lumbar vertebrae (L5) and on the left and right anterior-superior-iliac-spine (L/R-IAS). All 28 cameras were set to record at 100 Hz. The motion capture system was calibrated at the beginning of the measurement day. The calibration resulted in an expected average marker position error of 24 mm as calculated by the Qualisys calibration software. The recorded motion capture data was manually inspected and misidentified markers were fixed. Gaps in the marker trajectories smaller 80 ms were interpolated automatically. If larger parts of a trajectory could not be reliably tracked, the sections were removed from the analysis. All motion capture analysis was performed using QTM version 2019.1 and final trajectories were exported into the c3d format to be further analyzed using Python. Specifically, the trajectories of the heel and the toe markers were used in this manuscript.

Each participant performed seven gait tests: First, the participants performed three iterations of a 4×10m walk test at preferred, self-selected slow, and self-selected fast speeds. Then three iterations of a 2×20m walk tests were performed with the same speed levels as the 4×10m tests. Participants were instructed to always turn in the same global direction at the end of each 10/20 m segment. This resulted in an equal number of left and right turns in the dataset. These two tests were included to provide comparability with other datasets, which typically include at least one of these two tests. To collect a larger number of strides, the participants performed a continuous 5min-walk along a path shaped like an eight within a 20×5 m area. Participants started in the center of the eight and walked the first curvilinear section in a counterclockwise (left turn) direction. This path shape provides the two long diagonals for steady state walking and areas of curvilinear walking in both directions. However, only the straight walking portions are analyzed in this manuscript. At the beginning of the 5min-walk, participants were asked to start in their preferred speed. After two minutes, they were instructed to walk in their self-selected slow speed for another two minutes. In the final minute, the

participants walked in their self-selected fast speed. Due to the large capture volume, the complete gait tests could be covered with the reference system.

For one participant one of the sensors malfunctioned, and a second recording was made. Only the second recording is used in this analysis. Further, for three participants the cavity sensor was placed in the cavity the oriented the wrong way. This was fixed by rotating the data after loading. More information about this process can be found in the documentation provided with the dataset.

To reduce the amount of data generated by the motion capture system, the recording was started and stopped before and after each gait test. This means that seven recordings exist for each participant (one for each gait test). In contrast, the IMUs recorded continuously to their internal storage over all gait tests, resulting in only a single recording per sensor per participant.

All IMUs were synchronized with each other using the method introduced in [14]. The IMU network was in turn synchronized with the camera system by attaching a modified sensor unit to the analog synchronization output of the Qualisys camera system. Every time a recording was started with the Qualisys system the analog signal passed to the output switched from low to high and from high to low once the recording ended. This rectangular signal was recorded by the modified sensor and the rising and falling edges were then used to cut and align the recordings of the IMUs with the camera recording. This results in a synchronization between the two systems with an error smaller than 10 ms.

The study was conducted according to the guidelines of the Declaration of Helsinki, and approved by the local ethics committee of the Friedrich-Alexander University of Erlangen-Nürnberg, Germany (Re-No. 106_13B, 19/03/2020). All participants provided written consent to participate in the study and to have the recorded data published.

## Sensor alignment

In order to compare the raw signal of the sensors, a method was developed to optimally aligned their orientations. This method can extract the alignment information from the recording itself and removes the need for dedicated calibration motions, like the ones used in [23].

First, the signal around each gait test was extracted for all sensors on one foot as follows: At the beginning of each gait test, when the subject was instructed to stand still for a couple of seconds, a static region of 500 samples (2.4 s) was extracted with a sliding window approach. A window was considered static for a sensor if the variance of all accelerometer axes individually within the window were below 0.01 m/s$^2$. The first window where this was true for all sensors of one foot simultaneously was used to define the direction of gravity within each sensor using the median acceleration within the window. With that information, the coordinate systems of all sensor units were rotated so that the $z$-axis of the local coordinate system aligns with the direction of gravity during this resting period. As this step unifies the global $z$-direction in all sensors, the only remaining degree of freedom is the rotation around this $z$-axis. This alignment is performed in the second step using the measured angular velocity.

For the mathematical description, we approximate the shoe to be a solid object during most of the gait cycle. Therefore, we expect the recorded angular velocity of all sensor units to be approximately identical in the global coordinate system. This constraint can be used to derive the relative orientation of the local coordinate systems of two sensors. To provide an alignment of all sensors, we will use the approach described in the following to align all sensors individually with the cavity sensor. However, the procedure is identical for any arbitrary pair of two sensor units $i$ and $j$ attached to the same foot.

In general, to find the relative orientation for two sensors attached to a solid object, one would need to solve the following equation for the rotation matrix $R_{ij}$:

$$\boldsymbol{\omega}_j = \mathbf{R}_{ij}\boldsymbol{\omega}_i, \tag{1}$$

where $\boldsymbol{\omega}_{i,j}$ denotes the measured angular velocity of the sensors $i$ and $j$, respectively, and $\mathbf{R}_{ij}$ is the rotation matrix rotating the local coordinate system of sensor $i$ into the local coordinate system of sensor $j$. This equation has an infinite of solutions. In the following, we will ignore this fact, because every solution would solve the alignment problem, and always assume, we are looking for the shortest rotation between the two vectors. Even then, we would not expect the equation to provide the same rotation matrix at every point in time in the real world. Because of the expected noise of the sensor signal and the fact that the shoe will bend and deform during parts of the gait cycle, our initial assumption will be violated. In result, the matrix $\mathbf{R}_{ij}$ is actually time dependent. Because we want to use $\mathbf{R}_{ij}$ to perform an initial global alignment, what we are interested in, is an optimal solution for Eq 1 over the entire recording. This optimal rotation matrix could be found by means of a numerical optimization over multiple time points.

However, because we can already assume that the local $z$-axis of all sensors are aligned because of the gravity alignment, the problem can be reduced to one dimension. Instead of trying to find the entire rotation matrix, we only need to derive the angle $\Delta\varphi_{ij}$ that aligns the two sensor coordinate systems by rotation around the already aligned $z$-axis. We consider $\boldsymbol{\omega}_{xy,i}$ and $\boldsymbol{\omega}_{xy,j}$ to be the 2D vectors describing the angular velocity of the sensors $i$ and $j$ in the $xy$-plane. Adapting Eq 1 to this simplified 2D case, the following must hold:

$$\boldsymbol{\omega}_{xy,j} = \mathbf{R}\{\Delta\varphi_{ij}\}\boldsymbol{\omega}_{xy,i}, \tag{2}$$

where $\mathbf{R}\{\Delta\varphi_{ij}\}$ is the rotation matrix rotating a vector around the $z$-axis by $\Delta\varphi_{ij}$. In the $xy$-plane $\Delta\varphi_{ij}$ describes the angle between $\boldsymbol{\omega}_{xy,i}$ and $\boldsymbol{\omega}_{xy,j}$. This angle can be derived using basic trigonometry:

$$\Delta\varphi_{ij} = \mathrm{atan2}(|\boldsymbol{\omega}_{xy,i} \times \boldsymbol{\omega}_{xy,j}|, \ \boldsymbol{\omega}_{xy,i} \cdot \boldsymbol{\omega}_{xy,j}) \ . \tag{3}$$

As we do not assume this value to be the same for every point in time, we calculated the angle for every sample in the signal and used the median to find the best approximation for the alignment angle. To make this step more robust, we excluded all samples where $|\boldsymbol{\omega}_{xy}|$ of one or both sensors were below a threshold of 150 deg/s. For values smaller than the threshold, we assumed that the result was considerably impacted by measurement noise and therefore unreliable. The final calculated angle was then used to fully align the local coordinate systems of the sensors $i$ and $j$. The whole procedure is visualized in Fig 3.

For all further data analysis, this method was used to align all sensors of a shoe with the respective cavity sensor. This results in the alignment of all sensors, which allows to apply the same analysis pipeline for all sensors going forward. The cavity sensor was chosen as reference, because its orientation is clearly defined by the cavity manufactured into the mid-sole of the shoe. The alignment was performed per gait test to minimize the influence of potential shifts of the sensor attachments over the duration of the entire measurement. The quality of all alignments was checked manually and all parameters and thresholds were chosen empirically.

## Stride segmentation and event detection

Stride segmentation is a critical first step of any gait analysis pipeline. Because this paper focuses on the physical differences of the sensor signals recorded at different positions, we wanted to reduce potential errors caused by the stride segmentation. Therefore, all strides in

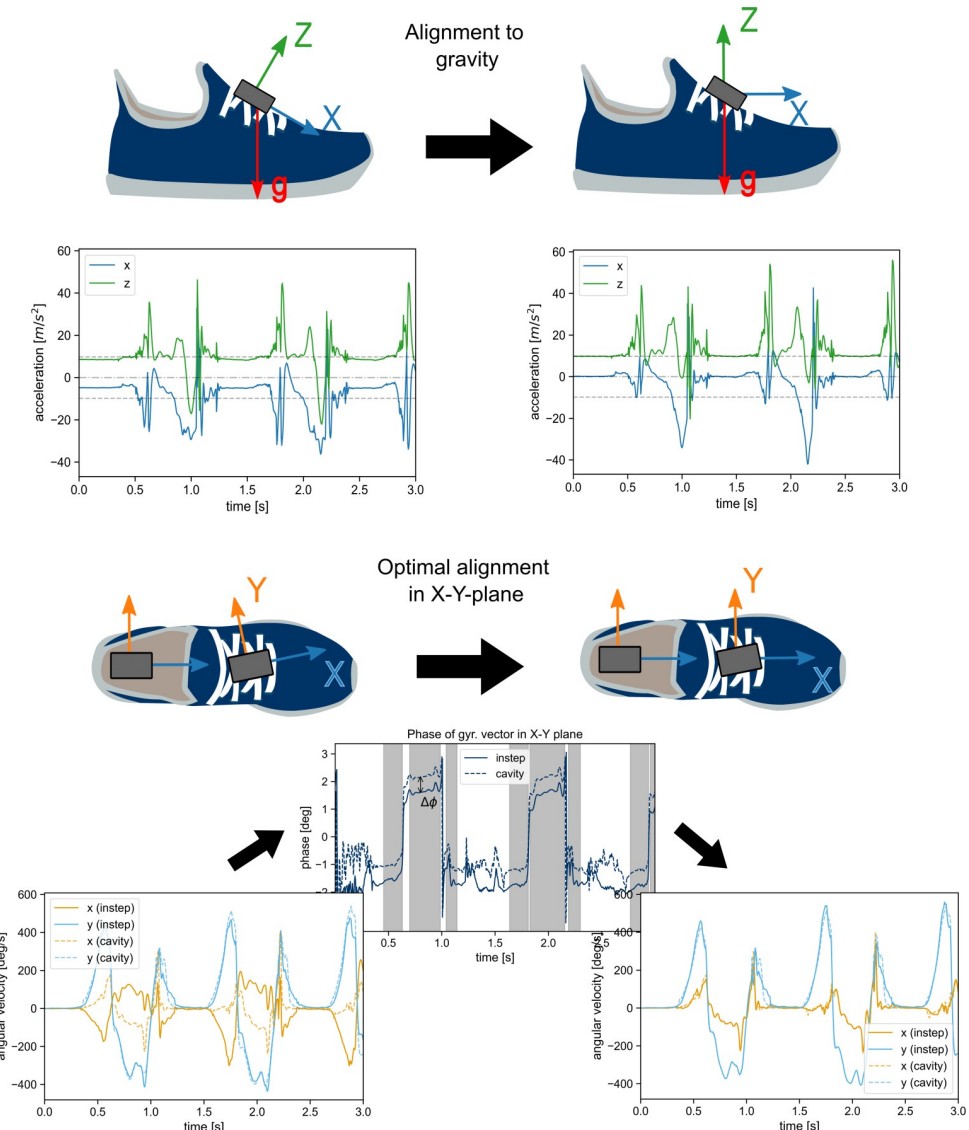

**Fig 3. Visual representation of the two-step alignment procedure.** The image shows the alignment of the instep sensor, but the same procedure is applied to all sensors. First, the local coordinate system is rotated, so that the z-axis aligns with the global direction of gravity during the foot-flat posture. In the second step, the rotation around the new z-axis is aligned with the cavity sensor by calculating the difference in phase between the angular velocity vectors $\boldsymbol{\omega}_{xy}$ in the x-y-plane. From the regions where the angular velocity is larger than 150 deg/s (gray areas in the plot) the median angle between the two sensors $\Delta\varphi$ is extracted to fully align the signals of the sensors.

the dataset were labeled manually by gait experts independently based on the medial-lateral gyroscope signal of the instep sensor. The start and the end of each stride were marked based on the clearly visible minimum before the terminal contact (for more information about the labeling process see [31]). To ensure consistency, the manual labels were moved to the exact sample of the minima in a 50 ms window around the labeled point. This ensured that the end label of one stride coincided with the start label of the next stride. This was further used to define adjacent strides and breaks in the gait sequence. A unique stride-id was assigned to each stride to reference the same stride in all sensor systems. In the following, we call these types of strides *labeled*-strides.

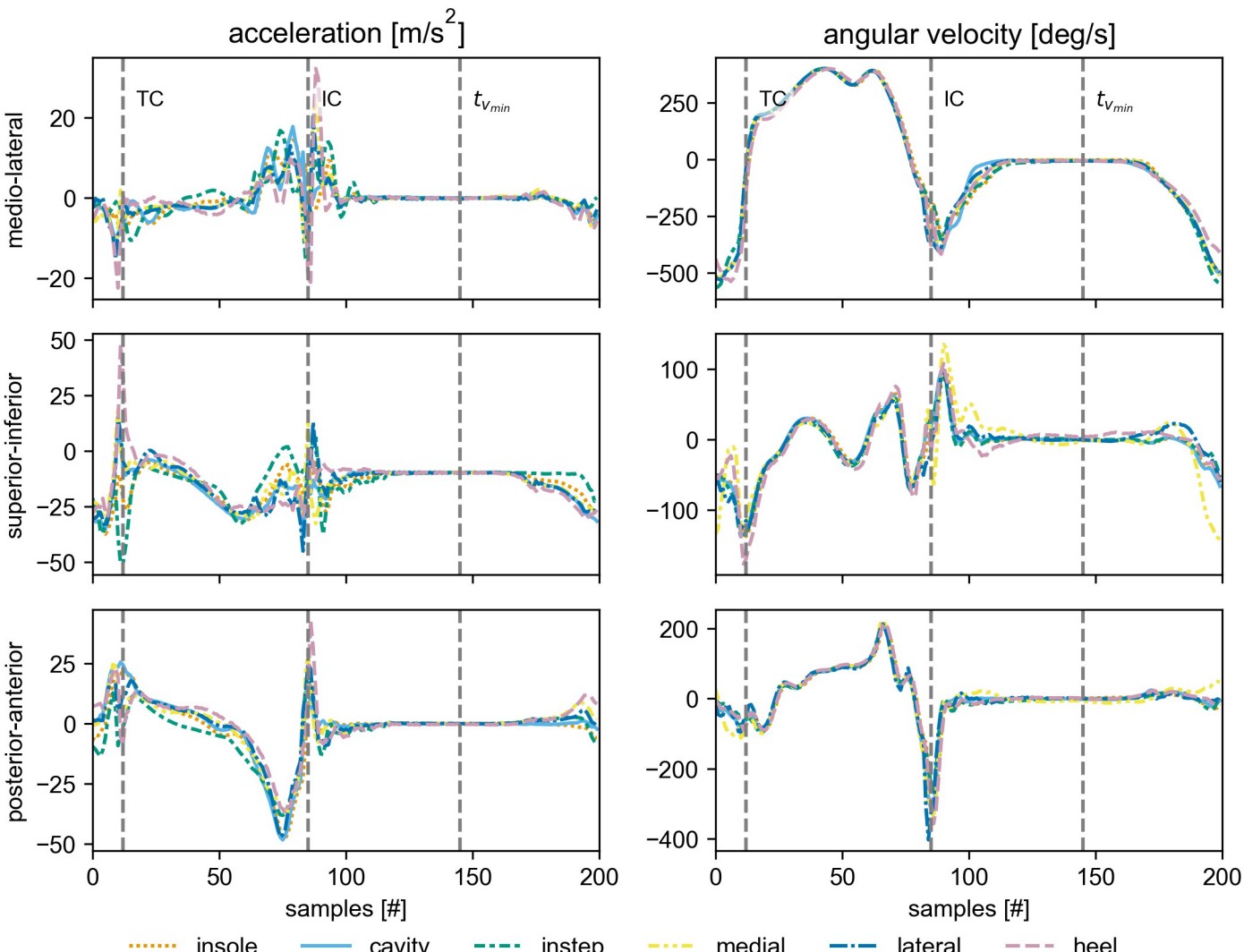

**Fig 4. Example stride.** The average signal over all strides of the left foot from the 2×20m walk test at normal speed of the participant 4d91. All strides were interpolated to 200 samples before averaging. The individual signals are rotated to match the primary anatomical axis during mid-stance. The start and the end of the stride conform with the *labeled*-stride definition. The vertical lines indicate the approximate position of the detected gait events.

This approach was chosen over calculating the stride events and stride borders from the motion-capture reference, as it was considered more reliable in our testing and allowed to label the strides in a way that was directly compatible with the used IMU-based event detection method in the next step.

The *labeled*-strides were then used as regions of interest to perform an event detection based on Rampp et al. [32]. This event detection was performed for each sensor separately. We detected the initial contact (IC), the terminal contact (TC), and the point of the least movement ($t_{v_{min}}$) during the stance phase (Fig 4). These events correspond to the events heel-strike, toe-off, and mid-stance in the original publication, respectively. As this manuscript focuses on the comparison of spatial parameters, the events were not used as part of the comparison later, but to further filter out strides that might contain signal artifacts (see section Stride selection for more information).

From the remaining strides we created a new stride list, with redefined start and end points. The new strides start at the $t_{v_{min}}$ of one stride and end at the $t_{v_{min}}$ of the adjacent stride. This ensures that the signal of each stride started with a resting period/stance phase, which allows to apply a stride-wise integration (see section Gait parameters). These new types of strides are referred to as $v_{min}$-strides. If there are breaks in the gait sequence, either due to actual breaks or removed strides, no direct adjacent stride exists based on the provided definition for adjacent strides. This leads to a loss of one stride per gait sequence compared to the *labeled* stride definition. A stride-id is assigned based on the stride-id of the *labeled*-stride the start $t_{v_{min}}$ belongs to. As the exact position of $t_{v_{min}}$ depends on the actual sensor signal, the start and end of each $v_{min}$-stride vary slightly per sensor. The stride-id is used to still allow a proper comparison of the same stride over multiple sensors.

## Gait parameters

This manuscript focuses solely on the estimated stride length for comparison. As it requires the estimation of the entire trajectory, the stride length is considered a quality marker for the entire calculation pipeline.

For calculation, we used a double integration approach similar to [23]. The initial orientation was estimated using a small window of 8 samples (40 ms) centered around $t_{v_{min}}$ at the start of each $v_{min}$-stride. We assume that the only measured acceleration in this window is gravity. Therefore, we assume that the median acceleration in the window aligns with the global $z$-axis. The $x$- and the $y$-axis are chosen orthogonally to this $z$-axis in a way that the $x$-axis aligns with the forward direction $((1, 0, 0))$ as much as possible. Starting from this initial orientation the global orientation of the sensor is derived by integrating the gyroscope measurements. With this global orientation, the measured acceleration is transformed into the global frame and the influence of gravity is removed. The remaining acceleration is then integrated twice, yielding the sensor position in the global frame. For the first integration a forward-backwards integration approach with sigmoidal weighting is used to remove potential drift. This method is chosen over the linear dedrifting in [23], as it has shown to provide the best results for normal walking [8]. This corrected velocity is then integrated again to yield the position. To derive the stride length, the traveled distance is derived as the Euclidean distance of the start and the end point of the integration in the ground plane ($x$-$y$-plane).

To obtain the stride length references from the motion capture recording, the position of the heel and the toe markers at the start and the end of each $v_{min}$-stride is detected. The reference stride length is then obtained by calculating the distance in the ground plane the markers traveled during the stride. While only the stride length based on the heel marker is used as actual reference going forward, the stride length from the toe marker is calculated as quality criteria (see section Stride selection). Because the event detection is performed for each sensor position separately, separate stride length references exist for each sensor. Their values vary slightly depending on the exact timing of $t_{v_{min}}$. This ensures that expected shifts in the event detection between the sensors do not influence the comparison.

## Stride selection

To ensure that all sensors were compared based on the same strides, all strides that had issues with the event detection in one or more sensors of the same foot were removed from the comparison based on the assigned stride-id. The event detection was considered failed, if the events were not detected in the expected order, or no IC was detected in the first 60% of the gait cycle.

Further, a fair comparison of different sensor positions is only possible for straight strides. If the orientation of the foot in the beginning of the stride is different from the orientation of the foot at the end of the stride, the actual traveled distance varies for different parts of the foot. This is a systematic error that also affects the comparison of different sensor positions and the comparison of markers and sensors that are attached at different positions on the foot.

To limit the influence of this error source, all strides that had a difference between the calculated reference stride length based on the heel and the toe marker of larger than 1 cm were removed from the comparison. This check was performed for the $v_{min}$-stride start and end values of all sensors individually and if the requirement was violated for one or more sensors of the same foot, the stride was removed. Because the stride length from the heel marker was used as actual reference, the largest remaining error because of this error source is expected for the instep and the insole sensors. Based on the selected threshold for straight strides, this remaining error is bound to be well below 1 cm. This approach has the side effect of removing strides that are affected by tracking issues of either the toe and the heel marker, too. Therefore, the remaining strides allow for a "best case" comparison of the different sensor position.

The final set of strides was further divided into six groups depending on the walking speed for the final analysis (Table 2).

## A physics model for the shoe-sensor system

To understand why we expect differences in the measured signal at different sensor positions, and hence, in the calculated spatiotemporal parameters, we need to establish a simplified mechanical description of the measurement chain. In the following section we attempt to derive such a model based on fundamental physics principals. This model is then used to hypothesize potential sources of error, which we attempt to investigate based on the available experimental data.

Fundamentally, we propose to differentiate between three types of signals that are measured independent of the sensor position: First, translations and rotations of the center of mass of the foot without microscopic or macroscopic deformations of the shoe or foot. Second, macroscopic deformations of the shoe, for example caused by the rollover of the foot before the terminal contact during a gait cycle. Third, microscopic deformations and the resulting pressure waves traveling through the material of the shoe and the foot. These are primarily caused by fast accelerations or deceleration, that can for example occur during the initial ground contact.

Depending on the type of signal, we expect different underlying sources of variation between individual sensor positions. For movements of the entire foot, we expect all sensors to measure the exact same angular velocity ($\boldsymbol{\omega}$). The component of acceleration $\mathbf{a}_s$ caused by the translation of the center of mass of the foot is also expected to be identically. Only the component of acceleration $\mathbf{a}_r$ caused by rotations (i.e., by the centripetal force) is expected to differ

**Table 2. Overview over the stride categories and the respective number of strides.**

|           | description                                                                                          | # strides |
|-----------|------------------------------------------------------------------------------------------------------|-----------|
| **slow**  | All strides from the 4×10m and 2×20m walk performed at the self selected **slow** speed.              | 1479      |
| **normal**| All strides from the 4×10m and 2×20m walk performed at the self selected **normal** speed.            | 1247      |
| **fast**  | All strides from the 4×10m and 2×20m walk performed at the self selected **fast** speed.              | 1043      |
| **straight** | The combination of all three speed categories.                                                    | 3769      |
| **5min-walk** | All strides from the 5min-walk.                                                                  | 5220      |
| **all**   | All strides from all test 7 tests (including the 5min-walk) combined independent of the gait speed.   | 8989      |

based on the sensor position, as it depends on the distance between the accelerometer and rotation axis:

$$\mathbf{a}_r = \boldsymbol{\omega} \times (\mathbf{r} \times \boldsymbol{\omega}), \tag{4}$$

where $\mathbf{r}$ is a position vector connecting the center of rotation and the sensor. Note that during gait, we have multiple additive rotational components caused by the simultaneous rotation around multiple joints (e.g., ankle and knee). This makes it hard to provide an actual estimate of magnitude of the rotational acceleration. To approximate the expected error range, we can make some rough assumptions: Assuming a maximal distance of 10 cm between two sensors in this experiment and a peak angular velocity of around 500 deg/s (approx. 8.5 rad/s), we can expect a maximum difference in rotational acceleration of around 0.75 m/s$^2$.

During gait, macroscopic deformations of the shoe primarily happen during the rollover of the foot between the foot-flat phase and the terminal contact. In this phase, the toe-box of the shoe still remains on the ground, while the heel is lifted. Naturally, sensors attached to the toe-box would measure a different signal compared to sensors attached to heel. In this experiment, no sensors were directly attached to the toe-box. However, the sensor placed in the insole and on the instep are placed close enough to the toes that they might be affected by the deformation. The primary expected influence on the sensor signal is a delay in the onset of the movement after the foot-flat phase. To a lesser extent, we expect macroscopic deformation after the initial contact (heel strike). Here we assume that sensors attached closer to the heel of the foot will stop their movement slightly before the sensors placed closer to the toe.

The influence of microscopic deformations and high velocity impacts (e.g., the heel strike) is the most complicated to predict, because it depends on the physical components involved in propagation of the forces from the point of impact to the actual sensor. This means the mechanical properties of the shoe, the foot, the sensor attachment, and the sensor itself will influence the measured signal. Simplified, each component will behave like a damped oscillator when it comes to the propagation of pressure waves caused by high accelerations and decelerations of the foot. The resulting system of such chained oscillators will result in a frequency dependent and—in the real world—likely non-linear (i.e., the frequency response will depend on the magnitude of the force) propagation of the pressure wave. The properties of this system will vary for each sensor position, because they have different means of attachment and the pressure wave needs to travel through different layers of material depending on the sensor position. For example, the sensor attached on the instep is expected to measure a lower peak acceleration after the initial ground contact compared to a sensor in the cavity, because the pressure wave is already damped and modulated by the shoe and the foot before reaching the sensor. Even at the same sensor position, the firmness of attachment might influence the amplitude of the measured signal. This is a well-known issue when estimating ground contact forces using accelerometers [33, p. 347-354].

Together all these effects are expected to result in clearly observable differences in the measured raw signal. However, based on presented theory, none of them would directly result in differences in spatial parameters like the stride length (other spatial parameters like contact angles might be affected). In this study spatial parameters are calculated by integrating the measured signal to obtain the overall spatial displacement of each sensor. Because the described differences in the raw signal are not measurement errors, but can be explained by actual differences in the movement paths (microscopic and macroscopic) of the sensors, we expect the same stride length when integrating these signals over one gait cycle of a straight stride, because all sensors need to travel the same distance in the ground plane between two foot-flat phases independent of their actual path during the remainder of the gait cycle.

However, we postulate two pathways how the described signal differences can lead to actual integration errors. The first one is based on the ZUPT. As described in [25], a period of no movement during a gait cycle is required for most integration methods to correct drifts. Further, this time period is used to estimate the initial orientation of the sensor at the beginning of each stride using the direction of gravity. The lower the amount of movement during this period, the more reliable both of these operations will be. This means, if the amount of residual movement is high, larger errors in spatial parameters are expected. Zrenner et al. [23] could show that the amount of residual movement varied for each sensor position and attributed this primarily to the macroscopic deformations of the shoe during running and the firmness of the attachment of each sensor. The latter aspect is one part to the frequency response of the entire measurement chain, as explained above. During regular gait we expect the frequency response to be the dominant factor, as we expect a lower amount of macroscopic deformation of the shoe compared to running.

The second potential source of error is caused by the limited sampling rate of the sensor system. In this study the sensors recorded with 204.8 Hz. This means that the highest theoretical signal frequency that can be recorded is around 100 Hz. This has shown to be sufficient to calculate the stride length with an acceptable error [5, 7]. However, acceleration measured during gait can contain components in frequency bands larger 150 Hz [34]. If this leads to relevant errors during the integration depends on the amount of signal energy in these higher frequency bands. Depending on the exact oscillation and dampening behavior of the measurement chain for each individual sensor, it is possible that some sensors will experience more movements in these high frequency ranges than others. As these movements cannot be correctly sampled with the sampling rate used in this study, there is missing signal information during the integration process, which can lead to errors in the orientation and position estimate that can compound over the integration process. Because such high frequency movements usually also result in high accelerations and angular velocities, this can lead to relevant integration errors, even though the actual magnitude of the movements might be small.

## Raw data analysis

While the comparison of the stride length is the primary analysis of the manuscript, it is of further interest how the raw signal itself is influenced by the change in sensor position. The aim of this analysis is contributing to a better understanding of the measurement response of a shoe attached IMU system and helping to pinpoint the exact origin of potential differences in spatial parameters that arise from differences in the frequency response. Further, the calculated parameters will be used to experimentally verify the error model outlined in the previous section. To achieve these aims we performed multiple comparisons of the raw sensor data between the sensor positions.

This analysis was split into two parts: First, we directly compared the raw sensor data sample-by-sample across all sensors and second, we calculated a set of parameters for each stride based on the raw data. These parameters were then compared between the sensors and correlated with the calculated stride length error.

**Direct raw data comparison.**   In addition to visually comparing individual strides and average signals over multiple strides, we attempted to capture the similarity of the raw sensor signals in quantifiable metrics. This was done using the *labeled* stride definition. This means each stride had the same start and end values across all sensors on the same foot, and hence, we can easily perform a sample-wise comparison. For this we derived two different summary metrics:

*3D Difference.* We calculated the sample-wise vector difference between two sensors of the angular velocity and the acceleration, respectively. We then calculated the norm of these difference vectors and averaged the result per stride. Hence, for each stride and sensor pair $i, j$ the following calculations were performed:

$$\Delta_{3D,acc} = \frac{1}{N} \sum_k |\mathbf{a}_{i,k} - \mathbf{a}_{j,k}|, \tag{5}$$

$$\Delta_{3D,gyr} = \frac{1}{N} \sum_k |\boldsymbol{\omega}_{i,k} - \boldsymbol{\omega}_{j,k}|, \tag{6}$$

where $N$ are the overall number of samples and $k$ the indices of the samples contained in a respective stride. The resulting *3D Difference* is per definition always positive and hence, cannot differentiate between noisy differences and actual data offsets.

*Norm difference.* Instead of deriving the difference of the raw signal per axis, we first obtained the vector norm of the angular velocity and the acceleration per signal and then calculated the sample-wise difference between sensors. The resulting value is then averaged over one stride:

$$\Delta_{norm,acc} = \frac{1}{N} \sum_k |\mathbf{a}_{i,k}| - |\mathbf{a}_{j,k}|, \tag{7}$$

$$\Delta_{norm,gyr} = \frac{1}{N} \sum_k |\boldsymbol{\omega}_{i,k}| - |\boldsymbol{\omega}_{j,k}|. \tag{8}$$

This difference metric can have negative values and is expected to be zero if the time-depended difference between two sensors has zero mean. Therefore, it can be used to differentiate between constant offsets of the signals and oscillations.

For both metrics we calculated the mean and interquartile range (IQR) over all available strides. Based on this we then identified sensor positions that result in similar signal (small differences) and tried to characterize the nature of the potential differences by comparing the *3D Difference* with the *Norm Difference* for individual sensor pairs.

Further, we analyzed when in the gait cycle differences between sensor positions occur. For this, we calculated the sample-wise standard deviation over all sensor positions per axis. We then interpolated the resulting values to 200 samples per stride and averaged over multiple strides for a qualitative comparison. A high standard deviation in a certain region of the stride indicates high disagreement between the sensor positions.

**Feature based data comparison.** To verify the postulated error model based on the collected experimental data, we calculated a set of features from each stride to see if they follow the expected patterns. To allow for comparison with the stride length result, we used the $v_{min}$-strides to calculate all parameters.

*Residual energy* ($E_{v_{min}}$). To estimate the residual movement during the stance phase, we calculated the energy of the gyroscope signal in a 40 ms window centered around the $t_{v_{min}}$ point that marks the start of each stride. Any value larger than zero indicates that the sensor was still moving during a time period we assumed to be a zero-velocity region. As the data in the same window is also used to estimate the initial orientation for the double integration and is further used to dedrift the calculated velocity, any residual energy is expected to negatively influence the stride length estimation in multiple ways. $E_{v_{min}}$ was calculated as the sum over the squared

norm values within the window:

$$E_{v_{min}} = \sum_i \|\boldsymbol{\omega}_i\|^2 \text{ for } i \in \{i \in \mathbb{N} | t_i \text{ with } t_{v_{min}} - 20 \text{ ms} < t_i < t_{v_{min}} + 20 \text{ ms}\}, \qquad (9)$$

where $t_i$ represent all the discrete sample-points within the described window, and $\omega_i$ the gyroscope sample at sample $i$.

*Peak acceleration* ($a_{max}$). As an indicator of the dampening of the shoe system we calculated the maximum of the acceleration norm in each stride. In most strides this maximum occurs right after the initial contact. The higher the damping effect of the specific measurement chain, the lower the expected signal.

*Power Spectral Density* (PSD$_{(0,20]}$/PSD$_{(80,102.4)}$). To better understand the different frequency responses of the measurement chain of each sensor, we calculated the power spectral density (PSD) of the accelerometer and gyroscope norm. Like in [34], we used Welch's method with a Hanning window of 64 samples and 50% overlap. We zero padded the windows of 64 samples to 128 samples to increase the spectral resolution of the resulting PSD. The calculation was performed using the `welch` function from the Python package *scipy* [35]. For quantitative comparisons we divided the spectrum into multiple sections and calculated the average spectral power in each region. In the context of this manuscript, we present the results for the lowest frequency bin (0–20 Hz) and the highest frequency bin (80–102.4 Hz). The former represents the frequency range of the actual human movement, while the latter represents high frequency components potentially linked to sampling related errors explained above. However, high spectral power in the high frequency bin does not directly mean that sampling issues will occur. By definition, these signals are still represented correctly in the sampled signal. However, high spectral density up until the Nyquist frequency might indicate that relevant signal components with even higher frequencies exist.

For each of the features, we further investigated if any direct correlation between the features and the stride length result could be observed by plotting the feature against the stride length error. We performed this analysis on multiple granularity scales: median values per sensor, median values per gait test, and the raw values per stride.

## Results and discussion

### Stride length error

**The absolute stride length error is dependent on the sensor position.** The final average stride length error per sensor position is the primary outcome of this study. When comparing the mean error over all strides between the different sensor positions, no relevant difference can be observed (Fig 5, right). All sensor positions result in mean errors of below 2 cm. This is comparable or even better than results presented in other studies (e.g., [6, 7, 32]).

Looking at the IQR (Fig 5) and the mean absolute error (MAE) (Fig 6, left), differences between the sensor positions become apparent. For these metrics the error increases in the following order: insole, cavity, instep, medial, lateral, heel. While the performance within the first three and within the last three sensors appears to be comparable, the performance differs between these two groups. However, for all sensor positions the performance values remain within the ranges reported in other studies.

To get further insight into the stride length error, the speed dependency of the error is investigated. For this the *slow*, *normal*, and *fast* stride groups are compared (Table 2). For all sensor positions, all error metrics increase with increasing speed (Fig 6, right). The aforementioned differences between the sensor positions can be observed at all speed levels, but become more pronounced at higher speeds.

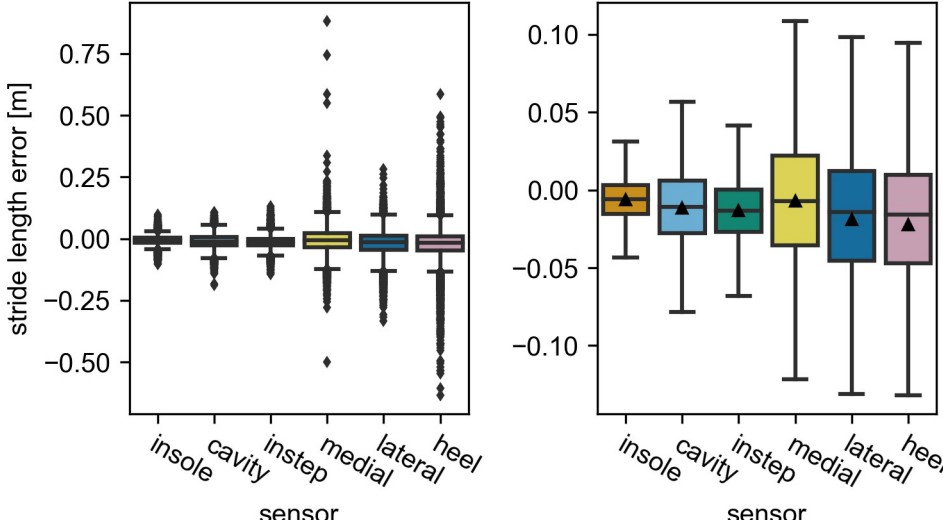

**Fig 5. The stride length error of each sensor position over all strides.** The image on the left shows all straight strides. The image on the right has all outliers ($> Q3 + 1.5IQR$ and $< Q1 − 1.5IQR$) removed for visual clarity. Values larger than 0 indicate an overestimation by the IMU. The black triangles mark the means of the distributions. They are omitted in the left image for visual clarity.

**The heel and the medial sensor have outliers with large errors.** Besides the MAE and the IQR, the number and spread of outliers vary between sensor positions and increase following the same order as the other metrics. This is notable, because these outliers can reach 60 *cm* for the heel sensor and even more for some strides of the medial sensor (Fig 5). This is well beyond an acceptable error range for any application and larger than the maximal errors reported in other studies. Unfortunately, no comparable study used the sensors at the affected positions. Therefore, no direct comparison is possible. For the remainder of this section, we will use the term *extreme outlier* for all strides with an absolute error larger than 30 cm.

For the medial sensor, only a handful of strides exceed this threshold. Otherwise, the error distribution is comparable to the lateral sensor. We assume that these extreme outliers are caused by participants occasionally bumping into the medial sensor with the opposite foot, which we observed during data recording. However, we could not find any clear indicator in the raw data to proof that this happened for the effected strides. For the heel sensor, the error distribution continuously spreads until an error of around 60 cm. This indicates that these outliers appear statistically because of the sensor position and not due to special events. With regard to the speed dependency of the outliers, basically all extreme outliers occur during fast walking (S1 Fig).

**Observed differences are unlikely to be caused by methodological errors.** To sanity check these results, we reran the stride length estimation using a Kalman Filter similar to the one used in [6] applied to the entire gait tests, instead of the per-stride double integration. We obtain comparable results with this alternative method and hence, concluded that the observed errors (including the outlier) were not caused by any issues with our integration method itself.

As explained in the methods section, the start and end time of the integration regions varied slightly per sensor. Because the same start and end values were used to obtain the reference stride length for the respective sensor, the reference values are also expected to vary slightly. However, tracking issues of the Motion Capture system might lead to larger differences than expected. To check for that, we subtracted the reference stride length for the insole sensor

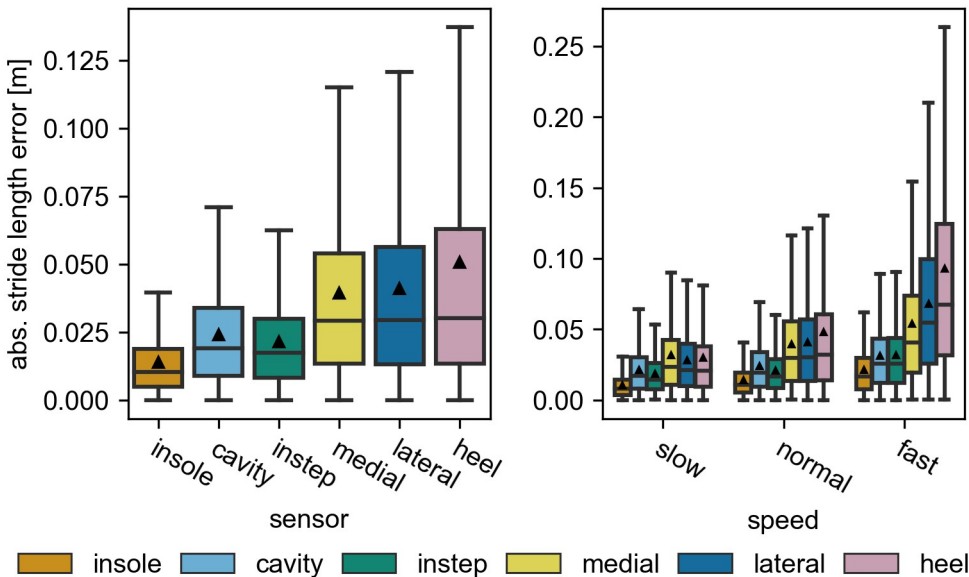

**Fig 6. The absolute stride length error of each sensor position.** On the left the error is calculated over all strides and right for the individual speed categories. All outliers ($> Q3 + 1.5IQR$) are removed from the visualization. The black triangles mark the means of the distributions.

from all other references per stride. The maximum difference for an individual stride was found to be 1.05 cm. Therefore, differences in the stride length reference or tracking issues of the motion capture system can be ruled out as a source of the large errors observed for certain sensors.

To better understand the consistency of the observed error values, we compared the results from the left and the right foot (S2 Fig) and the results of all participants individually (S3 Fig). No relevant difference between the results from the two feet could be observed. Between participants, the absolute values of the mean errors varied. However, the relative ranking of the sensor positions remained consistent for most participants. This rules out individual sensor units, calibration issues, or accidental operational errors as hidden confounders. We explain the remaining differences between participants by variations in walking style, small variations in sensor mounting, and differences in how well the shoes fit.

To summarize, while the average error over all strides is comparable between sensor positions, on a single stride level clear performance differences can be observed. This also means that measures for gait variability like the stride length standard deviation vary between sensors. Because the general error ranges of all sensors are comparable with other studies, and we controlled or tested for all obvious confounding factors, we are confident that the observed errors are representative for the specific sensor positions in our setup and not caused by systematic issues of the sensors or algorithms. Hence, the differences in stride length must be caused by actual differences in the raw IMU signal between the sensors.

### Raw data analysis

**The raw data of the sensors show clear differences.** In addition to the spatial parameters, we analyzed differences of the raw signal between the sensors. In the following we will discuss these differences qualitatively. All comparisons were performed over *all* strides (Table 2).

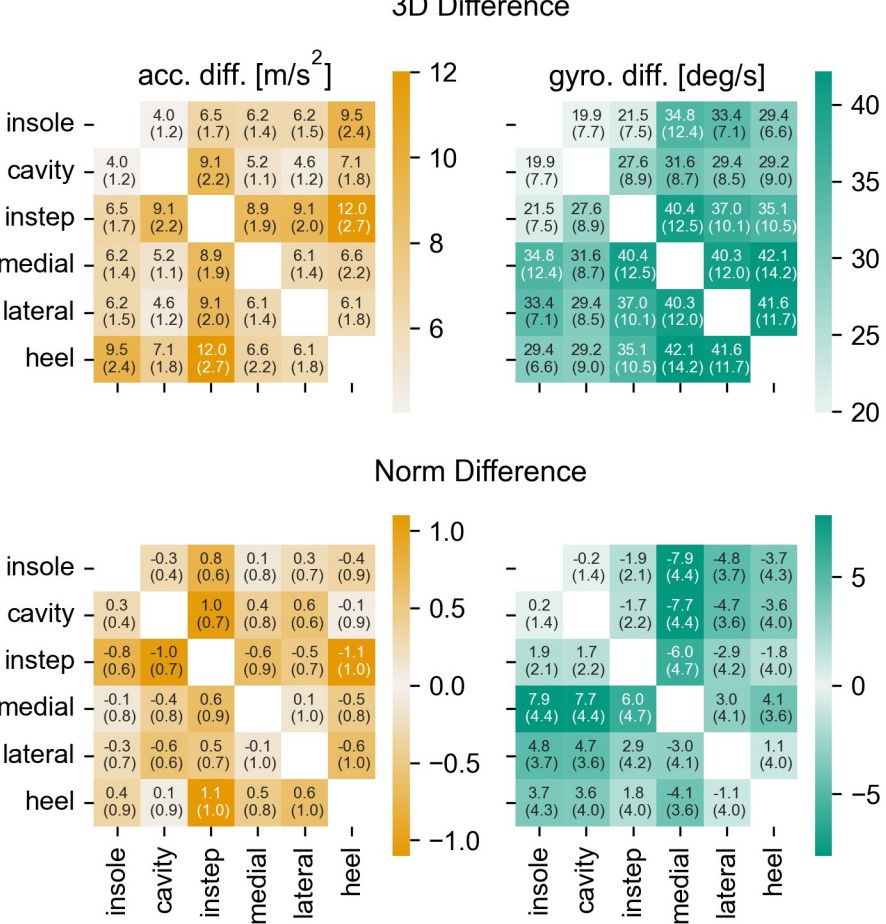

**Fig 7. The raw data differences for all sensor axes.** The *3D Difference* (top) and the *Norm Difference* (bottom) over all strides for the accelerometer signal (left) and the gyroscope signal (right). The annotations provide the mean and the IQR over all strides. Note that the color for the *Norm Difference* is independent of the sign of the value.

Looking at the *3D Difference* of the raw data (Fig 7), it is apparent that the insole and the cavity sensor are most similar. In contrast, the acceleration signal of the instep sensor appears to have the largest differences from all other sensors. For the sensors attached to the collars of the shoe (medial, lateral, heel) the angular velocity signal seem to vary considerably between the sensor positions. The first two observations can be explained based on the proposed physics model: The insole and the cavity sensor are rigidly connected and hence, are expected to measure the same angular velocity. The acceleration only differs by the difference in centrifugal acceleration between the sensor positions. This value should be relatively small, as the distance of both sensors to the involved joints is comparable. In contrast, due to the way the instep sensor is attached, it is likely that the sensor is affected by the macroscopic deformation of the shoe during rollover. Hence, it is expected that the movement of the instep sensor differs from the other sensors. This can be seen most clearly in the superior-inferior acceleration in the example stride (Fig 4). The instep sensor reaches a static phase slightly after all other sensors and starts moving slightly later. The observed difference between the sensors on the collar of the shoe challenge the assumption that the back portion of the shoe can be considered

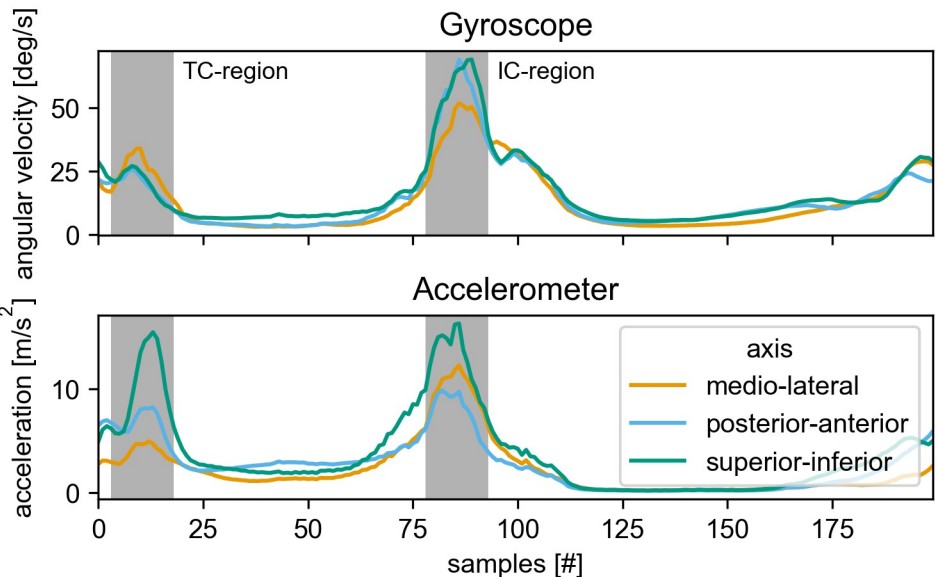

**Fig 8. The STD over the different sensors per sample of a stride.** The curves are the averages over all strides from the 2×20m walk test at normal speed. The STD of each stride was interpolated to 200 samples before averaging. The individual signals are rotated to match the primary anatomical axis during mid-stance. The approximate regions around the IC and TC are marked in gray.

quasi-rigid. It appears that either through the interaction with the foot or because of the flexibility of the collar, independent movement components exist in each of the three sensors.

Looking at the *Norm Difference* of the acceleration (Fig 7) the sensors on the collar appear to be much less different. This indicates that the high values for the *3D Difference* are caused by random signal variations that average out over multiple strides, rather than by a consistent offset between the signals. It can further be observed that the *Norm Difference* of the acceleration for the medial sensor with the instep, cavity, and insole sensors is considerably larger than for the other sensors on the collar. This points to a high-frequency "noise-like" signal component that is different between the sensors. For the remaining sensors, the *Norm Difference* only confirms observations that were already made based on the *3D Difference*.

When investigated further, it seems that the observed differences between the sensors are located in the regions around the IC and TC (Fig 8). For the acceleration the largest differences appear to be in the superior-inferior axis. The sensor signals during the mid-stance and the swing phase are highly comparable. This is in line with the proposed physics model, as differences between the sensors should only occur, if there is macroscopic deformation of the shoe, strong impacts, or fast movements. All of these only occur in the transition phases between stance and swing phase.

In summary, this raw data analysis shows that the measured raw signal between the sensors is qualitatively different. Further, the accelerometer and the gyroscope need to be considered independently. Just because two sensor positions result in a comparable signal in one of the sensor modalities, does not mean that they are similar in the other.

**The signal differences can be quantified using the calculated features.** To even better understand the nature of these differences, we analyzed specific features on the raw data. These features are calculated on the straight walking tests to directly visualize the speed dependency of the features. However, comparing the features on all strides leads to comparable conclusions with regard to the sensor position.

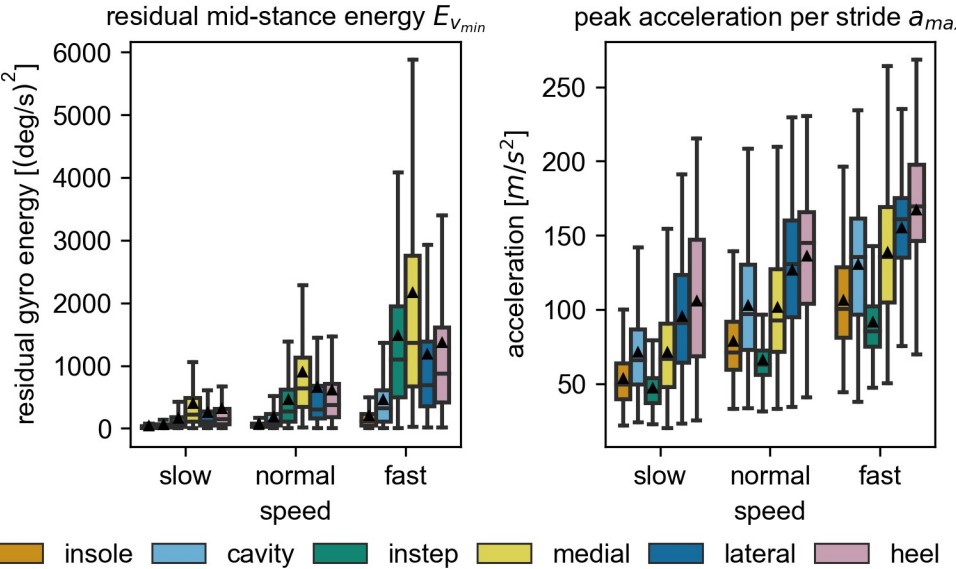

**Fig 9. The raw data features over all speeds.** The residual energy $E_{v_{min}}$ during the mid-stance is shown on the left and the peak acceleration $a_{max}$ during the gait cycle is shown on the right. Both boxplots contain all strides of each of the self-selected speed categories. Note the different y-scales for the individual plots. The black triangles mark the means of the distributions. All outliers ($> Q3 + 1.5IQR$) are removed from the visualization.

Looking at the residual energy $E_{v_{min}}$ during the stance phase, a clear position dependency can be seen (Fig 9, left). Sensors placed under the foot in the mid- and insole of the shoe have a relatively low mean residual energy independent of the gait speed. Sensors attached to the upper part of the foot have a higher amount of residual energy, indicating more movement during the mid-stance phase. Independent of the sensor position, the overall spread of values is large and for all sensor positions the residual movement reached zero for some strides. As expected, the average residual energy increases with an increase in gait speed. This increase is larger for the sensors attached on top of the shoe. The instep sensor seems to be most sensitive to this. Its mean residual energy has the largest relative increase going from the low to the fast speed category. Comparing the medial to the lateral sensor, which were expected to behave similarly, it can be observed that the residual energy of the medial sensor is higher at all gait speeds.

The maximum acceleration peak $a_{max}$ also varied considerably between sensor positions (Fig 9, right). Similar to the residual energy, the overall variation is large and the mean peak acceleration increases with gait speed. However, compared to the other metrics, the differences between the sensor positions are smaller for higher gait speeds. The smallest overall peak acceleration is experienced by the instep sensor followed by the sensor in the insole. The cavity and the medial sensor show comparable values for this parameter. The highest peak accelerations are measured in the lateral and heel sensor.

**The differences between the sensors are located in the high frequency bands.** Looking at the frequency based features (Fig 10), the comparison shows that the majority of signal energy is located in the low frequency bands ($PSD_{(0,20]}$). Based on the qualitative comparison, all sensors appear to behave very similar in the low frequency bands. However, a quantitative analysis shows slightly higher energy values for the lateral and heel sensor in the acceleration energy and the medial sensor in the gyroscope energy. As expected, the overall PSD scales with the gait speed.

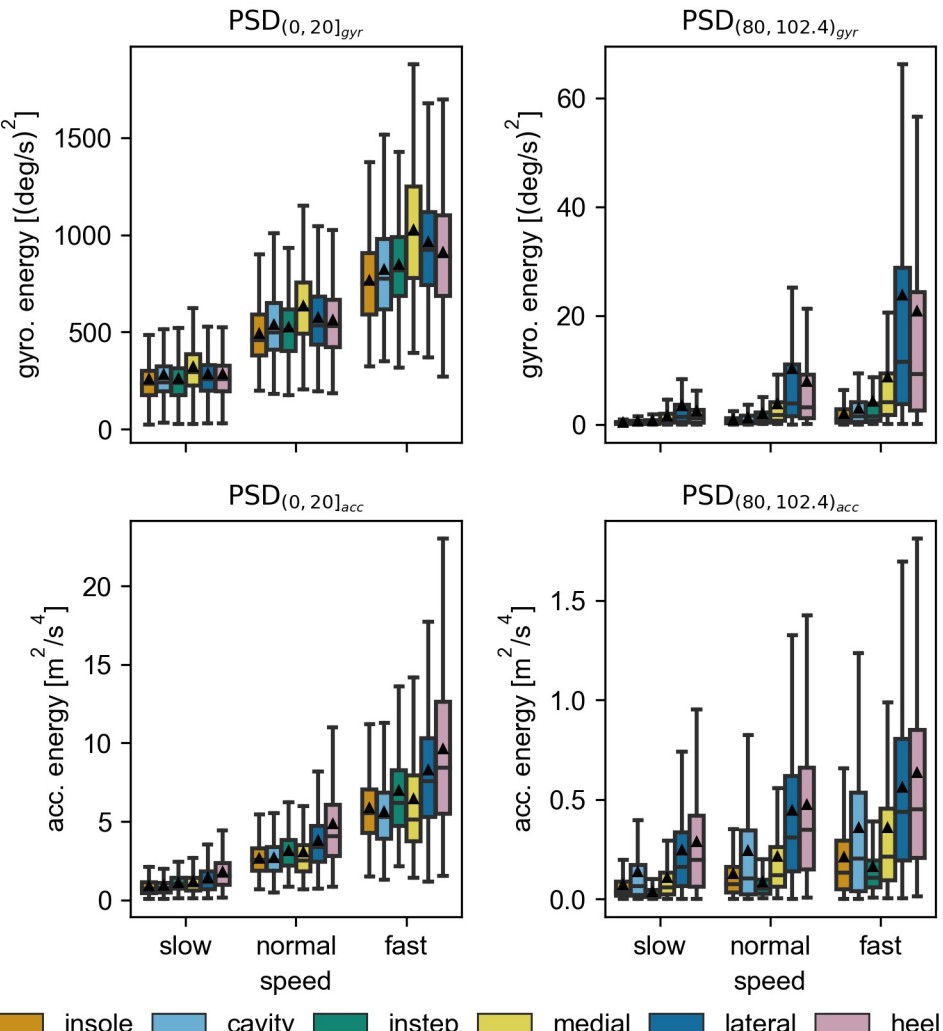

**Fig 10. The average power spectral density for the high and low frequency bins.** The average spectral power of the accelerometer (bottom row) and the gyroscope norm (top row) in the frequency ranges 0–20 Hz (left column) and 80–102.4 Hz (right column). Each datapoint used in the boxplots corresponds to a single stride. The black triangles mark the means of the distributions. All outliers ($> Q3 + 1.5 IQR$) are removed from the visualization.

In the higher frequency bands ($PSD_{(80,102.4)}$), clear differences between the sensors can be observed. In particular for the angular velocity, the measured average PSD for the lateral and heel sensors are considerably higher. This is most pronounced at the highest gait speed. The instep, the cavity, and the insole sensor all have similarly low values. For the acceleration the lateral and heel sensors again show the highest values. However, in contrast to the angular velocity, the cavity sensor results in next highest mean PSD with values comparable to the medial sensor. The insole and the instep sensor both have low spectral energy. Qualitative comparisons of these values between participants (not shown here) show that the exact order in this ranking depends on the participant, but stays consistent over all speed levels. For all participants either the lateral, the medial, or the heel sensor resulted in the highest measured mean PSD.

These differences in the calculated features underline that the sensor position influences the signal in multiple hard to predict ways. Therefore, it is likely that the calculation of most

kinetic and kinematic gait parameters is influenced by the sensor position, in particular when their calculation is based directly on raw data features.

**The proposed physics model can partially explain the measured differences.** With a good understanding of the position dependent differences in the raw data, we compared the results to the proposed physics model.

Starting with the frequency based features, the PSD results fit the proposed model for the physical properties of the shoe system and the sensor attachment for the most part. The sensors in the mid- and insole are tightly integrated in the shoe and are further held in place by the weight of the participant during the stance phase. Hence, we assume that the sensors and the surrounding part of the shoe will not be able to move a lot compared to the other sensor positions (i.e., the attachment is expected to have a high damping coefficient). In the measured parameters we can see that these sensors have low overall measured gyroscope energy as represented by the PSD and the lowest amount of residual movement during the mid-stance. The sensors on the upper part of the shoe can move more easily, because they are attached less rigidly and the parts of the shoe they are attached to are able to move and vibrate as well (i.e., the attachment is expected to have a low damping coefficient). Hence, we expect more movement in these sensors in particular right after the impact caused by the initial contact. This is reflected in the data by the overall higher mean PSD and higher residual movement during the mid-stance for the heel, the lateral, and the medial sensor. Only the instep sensor appears to follow a more complicated pattern, which might be explained by a combination of two effects. With regard to attachment, we would assume a similar potential for movement as for the other sensors mounted on the upper part of the shoe. However, because the sensor is placed far away from the heel, less initial movement might be introduced by the initial contact. At higher speed the force of impact is sufficient to introduce larger movements at this sensor position. This would then explain why the residual movement of the sensor is low at low speeds, but comparable with the other sensors mounted at the top of the shoe for higher speeds.

This is supported by the results of the peak acceleration. The measured value of the instep sensor is by far the lowest, supporting the assumption that the magnitude of the pressure wave reaching this sensor is reduced considerably due to the damping of the shoe and foot system. The insole sensor, which is placed similarly far away from the point of impact at the heel, has the second-lowest peak acceleration. However, comparing the cavity with the other remaining sensors, it becomes clear that the distance to the point of impact cannot be the only influencing factor. Otherwise, we would expect the cavity sensor to produce the highest signal. Hence, other parameters of the measurement chain must have an influence as well.

What cannot be explained by the model are the observed differences between the medial and the lateral sensor. Based on their comparable position relative to the heel and identical attachment method, the simplified physics model would suggest similar behavior. However, the results show differences between the two sensors for almost all metrics. This is even more surprising because the calculated stride length errors are highly comparable between the two sensors. Because the results of the lateral sensor are mostly comparable to the heel sensor, it seems to be the medial sensor that does not behave as expected. A possible explanation for this could be that healthy walkers are expected to roll off over the lateral side of the foot. This would cause more pressure of the foot against the lateral side of the shoe, which could change the frequency response of the sensor attachment. However, detailed investigations would need to be conducted to confirm this.

In summary, the proposed model is a good approximation to guide general intuition. However, the model is unfeasible to reason about the microscopic movements of the sensors as they depend on the actual frequency response of each component in the measurement chain.

To actually model this behavior correctly a much better understanding of the material layers and their interaction would be required.

**No simple correlation exists between the raw data and the stride length error.** Given the differences observed in the raw data between the sensors we investigated if any of the calculated features correlate with the stride length error. As explained in the methods section, we postulate two potential pathways for the observed differences in the stride length error: unreliable ZUPT measurements and signal components outside the measurable frequency spectrum. The first pathway can be directly assessed using the $E_{v_{min}}$ feature. However, we can not quantify the second pathway directly, as by definition, we can not measure the high frequency components that we assume to be correlated with the errors. Still, some features might be linked to the existence of these components. In both scenarios, the existence of any correlation between one of the raw signal features and the stride length error would not only help to better understand the origins of the errors, but could also be used a method to detect strides that might have high error values in real world applications.

A first comparison is performed based on the overall median values per sensor, comparing the stride length error (Fig 6) with the raw data features (Figs 9 and 10). The residual energy $E_{v_{min}}$ shows similar relative differences between the embedded sensors (cavity and insole) and the sensors attached to the collar of the shoe (heel, lateral, medial) as the stride length error. However, the stride length error of the instep sensor does not seem to follow its residual energy. This rules out ZUPT errors as the primary mechanism for stride length errors. Compared to [23], who postulated insufficient ZUPT updates as a primary source of position dependent differences in running, it appears that because of the absence of high movement speeds and large macroscopic deformations of the shoe in our study, residual movement during ZUPT only has a small (if any at all) influence on the stride length error. For the peak acceleration $a_{max}$, the sensors at the collar of the shoe have higher values than the embedded sensors. However, on closer inspection the scaling with the walking speed and the behavior of the instep sensor do not fit the stride length error. Hence, $a_{max}$ does not appear to be a reliable predictor of stride length error, too.

For the frequency based features, we hypothesized that large high frequency components close to the Nyquist rate ($PSD_{(80,102.4)}$) indicate that there are further signal components beyond the effective sampling rate of the sensor. If this is true a correlation between the signal energy in these frequency ranges and the stride length error is expected based on our error model. The $PSD_{(80,102.4)_{acc}}$ seems to follow a similar trend as $a_{max}$ and hence, does not correlate well with the stride length error. The median $PSD_{(80,102.4)_{gyr}}$ per sensor appears to capture the relative stride length error quite well. However, when directly comparing the values on a gait test or stride level (see Fig 11), the correlation is poor and has minimal predictive power. If this is because there is no correlation between the $PSD_{(80,102.4)_{gyr}}$ and frequency components beyond the Nyquist rate, or signal components beyond the Nyquist rate are not the origin of the observed differences in the stride length error, can not be assessed based on this dataset.

In summary, none of the calculated raw data features seems to strongly correlate with the stride length error. However, we could rule out ZUPT related issues as one of the mature sources of the observed differences. Beyond that, it is not possible to draw any definite conclusions. It is unclear, if simply no observable correlation exists between the raw data and the stride length error, or if multiple sources of error overlap and more complex models would be required to correlate the raw data with the stride length error. Multimodal prediction of the stride length error based on multiple raw data features might yield better results. Future work could explore this further with the goal of developing a continuous quality control for the calculated spatial parameters.

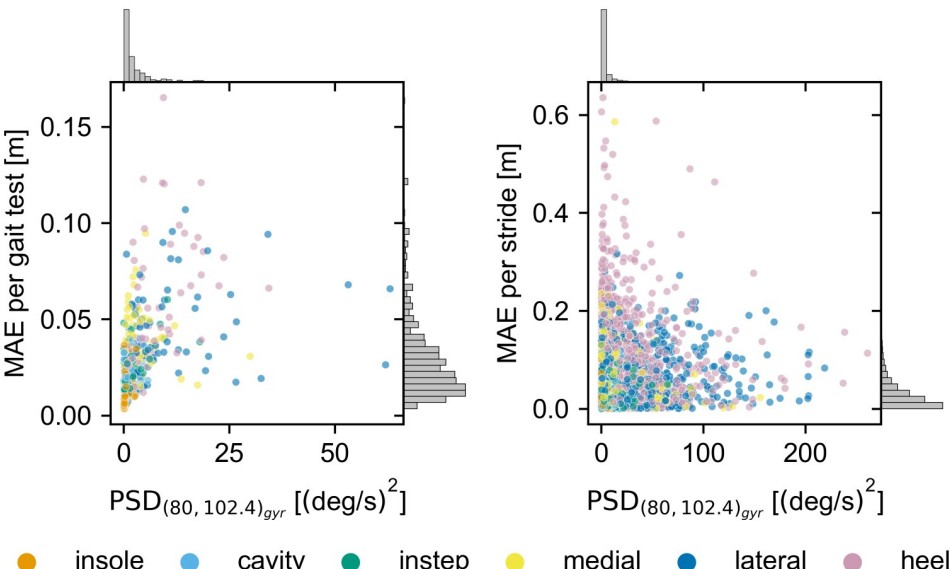

**Fig 11. The correlation between PSD$_{(80,102.4)_{gyr}}$ and the absolute stride length error.** On the left the correlation on a gait test level is shown. For that the median PSD and stride length error for each of the six straight walk tests are calculated per participant and sensor. On the right the correlation on a stride level is shown for the same tests. The histograms in the margins show the relative distribution of values across all sensors along the respective axis.

## General discussion

Compared to literature, we can replicate the result of [23], showing that the sensor placed below the foot results in the most reliable mid-stance. However, our observation differs for the instep sensor. The presented values for the residual energy of the instep sensor showed relatively low values. Zrenner et al. showed that the instep sensor has the highest amount of movement during the mid-stance. Even though the used metrics to quantify the residual movement were not identical, we would still assume comparable relative values between sensors. The differences for the instep sensor might be explained by the different biomechanics between regular walking and running. Further, in our data the residual energy of the instep sensor seemed to be highly sensitive to gait speed. Hence, the higher movement speeds during running might induce disproportionately more vibrations and movements in the instep sensor compared to other sensor positions.

While our results appear to compare well with the work of Zrenner et al., they seem to contradict the main conclusions from Peruzzi et al. [25]. Their data showed that virtual sensors (represented by motion capture markers) placed at the lateral side or the heel, experience the least amount of motion during the mid-stance and further have the smallest expected stride length error. In our experiments, it was exactly these sensors that showed the highest residual movement and resulted in the highest stride length errors. It is difficult to say, why we could not confirm the results from [25]. One explanation could be that motion capture markers cannot accurately simulate the behavior of a real IMU regarding these measurements. Further, Peruzzi et al. measured the lowest velocity of the marker during the stance phase, while we quantified the movement during the mid-stance via the gyroscope energy window. We could further show that the residual movement during the mid-stance is unlikely to be the source of error at regular walking speeds.

Compared to [22], we could confirm that the sensor position has a relevant influence on the calculated spatial parameter. However, we did not observe a large difference in the mean

stride length error, where Anwary et al. reported difference up to 6 cm in the calculated mean stride length. If this is due to the different algorithms used in our and their studies or caused by the differences in sensors and sensor attachments is unclear. Regarding the recommended sensor positions our results again seem to contradict the findings in [22]. Their results indicate that the heel is the second-best sensor position to calculate the overall walking distance. In our study, the heel position performed the worst. The most likely explanation is again the differences in attachment. Anwary et al. performed all of their experiments barefoot and used rubber straps to attach the sensors directly to the foot. Based on the presented simplified physics model this is expected to result in a very different frequency response of the measurement system compared to our study.

Our results add further proof that the sensor position matters when it comes to the quality of spatial parameters calculated from foot-worn sensors. While average values over multiple strides are expected to provide excellent results independent of the position, single stride parameters and therefore, parameters quantifying the variation of gait (e.g., stride length standard deviation) can yield unreliable values dependent on the sensor setup. The lack of agreement between all studies when it comes to the best performing sensor positions indicates that the sensor position alone cannot explain the observed differences. Based on our attempt to explain the physics behind the observed results it appears plausible that the entire frequency response of the measurement chain influences the quality of the final parameters. Hence, different shoes, different modes of sensor attachment, or how well a shoe fits might all be relevant factors that need to be considered. This makes comparing results from different studies difficult and adds multiple new parameters that need to be considered when developing new IMU based systems. While the presented simple physics model appears to explain some of the observed differences, it cannot capture the full range of parameters influencing the performance of the system in the real world. Therefore, further investigations into the topic are necessary.

For future work we see two general directions: On the one hand, a better understanding of the exact origins of the observed differences would be beneficial. To achieve this, a more systematic evaluation of the measurement chain might be necessary. This could include building measurement models at different levels of abstraction to understand what aspects of the measurement chain have relevant influence in a highly controlled environment. A different approach could be measurements with higher sampling rates than 204.8 Hz. If the observed errors are indeed caused by high-frequency components that cannot be correctly sampled at typically used sampling rates, high frequency recordings should be able to confirm this.

On the other hand, these findings open a range of application-oriented challenges that have to be addressed. Even without full understanding of the exact cause of the performance differences, it might be possible to develop algorithms that are less sensitive to changes in attachment. In particular, it would be interesting to see if data-driven methods based on machine learning, for example the algorithms presented in [36–38], are similarly sensitive to a change in sensor position as double-integration methods. Going beyond pure physics based calculation methods for spatial parameters might be a way to circumvent the described issues altogether, given sufficient training data for the individual attachment conditions. If it is not possible to improve the robustness of algorithms, it might be worthwhile to investigate methods for continuous quality control. In particular with home monitoring in mind, systems that are able to check, if a sensor is attached properly, at the correct position, and using a type of shoe that is expected to produce sufficient results could improve the quality of parameters extracted from such unsupervised scenarios. Combined with a better fundamental understanding of the sources of error, it might be possible to find indicators in the raw signal that are linked to large integration errors. Such features could serve as a continuous quality control

during long-term monitoring and could provide additional information when interpreting results.

## Conclusion

In this manuscript we present the most comprehensive study to date investigating the influence of the sensor position on the foot on spatial parameters calculated by double integration. We could show that the sensor position has a relevant influence on the accuracy of the stride length and could provide first evidence that the frequency response of the entire measurement chain can influence the final results in a relevant manner. Because our study could not control for the multitude of parameters that effect the frequency response, recommendations regarding sensor positions or attachments from this study (and other studies) might not carry over to other shoe or attachment systems. Therefore, we do not want to use this manuscript to provide a final recommendation, but rather urge researchers not to underestimate the influence of sensor position on the foot and in more general terms, the influence of the frequency response of the measurement system on the validity of spatial parameters calculated via double-integration methods. When the position of the sensor, its mode of attachment, or the type of shoe is changed, a proper reevaluation of the system must be performed to confirm expected error ranges. For home monitoring our findings further amplify the need for systems that are able to continuously check the quality of sensor recordings to avoid negative influences from user error or simply the variations introduced by regular behavior, like the use of different shoes. The implementation of such systems and the development of algorithms that are more robust to changes in the sensor position and sensor attachment parameters might be the key to robust and unsupervised long-term recordings.

## Supporting information

**S1 Fig. Absolut stride length error with outliers.** The absolute stride length error over all sensors including all outliers (compare Fig 6).
(TIF)

**S2 Fig. Consistency of the stride length results between the feet.** To check if the obtained results are consistent, we calculated the average stride length errors for each foot independently. The figure shows the absolute stride length error over all strides compared between the left and the right foot. The black triangles mark the means of the distributions. All outliers ($> Q3 + 1.5IQR$) are removed from the visualization.
(TIF)

**S3 Fig. Consistency of the stride length results over participants.** To check if the obtained results are consistent, we calculated the average stride length errors for each participant independently. The figure shows the absolute stride length error over all strides compared over all participants. All outliers ($> Q3 + 1.5IQR$) are removed from the visualization. The participant id *6dbe* actually refers to the recording *6dbe_2* in the published dataset.
(TIF)

**S1 Graphical abstract.**
(TIF)

## Acknowledgments

We thank the team of the Fraunhofer Institute for Integrated Circuits IIS for their measurement support in the L.I.N.K. test and application center. In particular, we would like to thank

Andreas Porada and Nicolas Witt for organizational and technical support before and during the data collection.

## Author Contributions

**Conceptualization:** Arne Küderle, Nils Roth, Bjoern Eskofier, Felix Kluge.

**Data curation:** Arne Küderle, Nils Roth, Jovana Zlatanovic.

**Formal analysis:** Arne Küderle.

**Funding acquisition:** Bjoern Eskofier, Felix Kluge.

**Investigation:** Arne Küderle, Jovana Zlatanovic.

**Methodology:** Arne Küderle, Nils Roth, Jovana Zlatanovic, Markus Zrenner.

**Software:** Arne Küderle.

**Supervision:** Arne Küderle, Bjoern Eskofier, Felix Kluge.

**Validation:** Arne Küderle.

**Visualization:** Arne Küderle.

**Writing – original draft:** Arne Küderle.

**Writing – review & editing:** Arne Küderle, Nils Roth, Jovana Zlatanovic, Markus Zrenner, Bjoern Eskofier, Felix Kluge.

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
