## [Decision Letter · Decision Letter 0]

7 Apr 2022

PONE-D-21-39063The placement of foot-mounted IMU sensors does affect the accuracy of spatial parameters during regular walkingPLOS ONE

Dear Dr. Küderle,

Thank you for submitting your manuscript to PLOS ONE. After careful consideration, we feel that it has merit but does not fully meet PLOS ONE’s publication criteria as it currently stands. Therefore, we invite you to submit a revised version of the manuscript that addresses the points raised during the review process.

We look forward to receiving your revised manuscript.

Kind regards,

YunJu Lee

Academic Editor

PLOS ONE

Journal Requirements:

“This work was supported by the Mobilise-D project that has received funding from the Innovative Medicines Initiative 2 Joint Undertaking (JU) under grant agreement No. 820820.

This JU receives support from the European Union’s Horizon 2020 research and innovation program and the European Federation of Pharmaceutical Industries and Associations (EFPIA).

Content in this publication reflects the authors’ view and neither IMI nor the European Union, EFPIA, or any Associated Partners are responsible for any use that may be made of the information contained herein.

This work was partially funded by the Bavarian Ministry for Economy, Regional Development & Energy via the Medical Valley Award 2017 (FallRiskPD Project).

This work was partially funded by the Deutsche Forschungsgemeinschaft (DFG, German Research Foundation) via the project "Mobility_APP" (grant number 438496663).

B. M. Eskofier gratefully acknowledges the support of the German Research Foundation (DFG) within the framework of the Heisenberg professorship program (grant number ES 434/8-1).”

Additional Editor Comments (if provided):

Due to the difficulty of finding the potential reviewers, please see the first reviewer's comment and edit the manuscript accordingly.

We might send more reviewers for your revision submission.

Appreciate your understanding.

Reviewers' comments:

Reviewer's Responses to Questions

**Comments to the Author**

1. Is the manuscript technically sound, and do the data support the conclusions?

Reviewer #1: Yes

2. Has the statistical analysis been performed appropriately and rigorously? 

Reviewer #1: Yes

3. Have the authors made all data underlying the findings in their manuscript fully available?

Reviewer #1: Yes

4. Is the manuscript presented in an intelligible fashion and written in standard English?

Reviewer #1: Yes

5. Review Comments to the Author

Reviewer #1: It is a well-structured article that focuses on IMU placement. However, a few parts needed to be clarified.

1. The background introduction and the idea of applying the existing devices to understand how the placement might affect the accuracy are exciting and practical. The purposes of the research are clear and applicable in many fields.

2. In the paragraph of “dataset,” the unit of gravity should use “G” instead of “g” (Line 115)

3. In the procedure part of the “dataset” (Line 148-163), “… a continuous 5 min-walk along a path shaped like an eight within a 20 x 5 m area.” Please describe if the participants walk in clockwise or counterclockwise. Or the participants could choose whatever they like? The purpose of this is that it might affect the walking pattern and might alter the medial or later placement IMU.

4. The data processing is sound, and the analysis of the data seems reasonable. However, the inclusion of other data (the residual energy, the peak acceleration, and PSD) might not be clear for readers in the following discussions. For instance, the main parameter of this research is stride length, which is the distance of the same foot at the minimum velocity at the closest different time points. Even though the maximum acceleration might occur after the initial contact, the correlation of the acceleration and the stride length was not clear enough for readers to understand. Similar to the PSD that higher spectral power in the high-frequency bin does not directly mean that sampling issues will occur. Please highlight the value of those parameters related to the placement of the IMU or the main parameters of strides.

5. If there is no significant difference between different conditions, please also provide analysis results that would help the readers understand better during the following discussion.

Thank you for your work and looking forward to more detailed sharing.

6. PLOS authors have the option to publish the peer review history of their article (what does this mean?). If published, this will include your full peer review and any attached files.

Reviewer #1: No

---

## [Author Response · Author response to Decision Letter 0]

25 Apr 2022

All comments raised by the editor and reviewer are addressed in the rebuttal letter titled "Response to Reviewers.pdf". The changes made to the manuscript are highlighted in the "Revised Manuscript with Track Changes.pdf" document

---

## [Editor Report · Decision Letter 1]

24 May 2022

The placement of foot-mounted IMU sensors does affect the accuracy of spatial parameters during regular walking

PONE-D-21-39063R1

Dear Dr. Küderle,

We’re pleased to inform you that your manuscript has been judged scientifically suitable for publication and will be formally accepted for publication once it meets all outstanding technical requirements.

Kind regards,

YunJu Lee

Academic Editor

PLOS ONE

---

## [Editor Report · Acceptance letter]

27 May 2022

PONE-D-21-39063R1 

The placement of foot-mounted IMU sensors does affect the accuracy of spatial parameters during regular walking 

Dear Dr. Küderle:

I'm pleased to inform you that your manuscript has been deemed suitable for publication in PLOS ONE. Congratulations! Your manuscript is now with our production department. 

Kind regards, 

on behalf of

Dr. YunJu Lee 

Academic Editor

PLOS ONE